# Defining the Patient with Lumbar Discogenic Pain: Real-World Implications for Diagnosis and Effective Clinical Management

**DOI:** 10.3390/jpm13050821

**Published:** 2023-05-12

**Authors:** Morgan P. Lorio, Douglas P. Beall, Aaron K. Calodney, Kai-Uwe Lewandrowski, Jon E. Block, Nagy Mekhail

**Affiliations:** 1Advanced Orthopedics, 499 E. Central Pkwy., Ste. 130, Altamonte Springs, FL 32701, USA; mloriomd@gmail.com; 2Clinical Radiology of Oklahoma, 1800 S. Renaissance Blvd., Ste. 110, Edmond, OK 73013, USA; db@clinrad.org; 3Precision Spine Care, 2737 S. Broadway, Tyler, TX 75701, USA; aaroncalodney@me.com; 4Center for Advanced Spine Care of Southern Arizona, 4787 E. Camp Lowell Drive, Tucson, AZ 85712, USA; business@tucsonspine.com; 5Independent Consultant, 2210 Jackson Street, Ste. 401, San Francisco, CA 94115, USA; 6Cleveland Clinic, 9500 Euclid Avenue, Cleveland, OH 44195, USA; mekhain@ccf.org

**Keywords:** lumbar, discogenic, pain, referred, diagnosis, reimbursement

## Abstract

There is an enormous body of literature that has identified the intervertebral disc as a potent pain generator. However, with regard to lumbar degenerative disc disease, the specific diagnostic criteria lack clarity and fail to capture the primary components which include axial midline low back pain with or without non-radicular/non-sciatic referred leg pain in a sclerotomal distribution. In fact, there is no specific ICD-10-CM diagnostic code to classify and define discogenic pain as a unique source of pain distinct from other recognized sources of chronic low back pain including facetogenic, neurocompressive including herniation and/or stenosis, sacroiliac, vertebrogenic, and psychogenic. All of these other sources have well-defined ICD-10-CM codes. Corresponding codes for discogenic pain remain absent from the diagnostic coding vernacular. The International Society for the Advancement of Spine Surgery (ISASS) has proposed a modernization of ICD-10-CM codes to specifically define pain associated with lumbar and lumbosacral degenerative disc disease. The proposed codes would also allow the pain to be characterized by location: lumbar region only, leg only, or both. Successful implementation of these codes would benefit both physicians and payers in distinguishing, tracking, and improving algorithms and treatments for discogenic pain associated with intervertebral disc degeneration.

## 1. Introduction

The intervertebral disc is the most intensively studied and characterized structure in the human spine [1,2,3,4,5,6,7,8,9,10,11,12,13,14,15,16,17,18,19,20,21,22]. Despite the enormous body of literature demonstrating that disc tissues are a potent source of experimentally induced low back pain [23], for almost a century, the concept of discogenic pain as a distinct clinical entity has not been well accepted [4]. In addition to creating definitional confusion among physicians and their patients, this lack of clarity about what constitutes symptomatic disc degeneration has had real-world implications in the way back-pain-associated conditions are defined and classified from a health information management perspective [1,24]. In simple terms, there are no International Classification of Diseases (ICD-10-CM) coding sub-terms for “discogenic” back pain.

In contrast, every other putative source of chronic low back pain, including facetogenic, neurocompressive including herniation and/or stenosis, sacroiliac, vertebrogenic, and psychogenic, has well-defined ICD-10-CM codes [25]. The lack of a corresponding code specifically for discogenic pain has hampered our ability to apply consistently accurate coding practices and nomenclature that yield quality data as dictated by the American Health Information Management Association (AHIMA) [26].

At one point in life, almost every individual will experience a sufficiently severe episode of low back pain to disrupt normal daily activities [27]. Fortunately, almost all of these patients will recover completely with time and conservative care [28]. Even recalcitrant cases of severe back pain respond reasonably well to intensive nonoperative multidisciplinary management [29]. Approximately 5% of patients, however, will suffer chronic, severe pain and functional impairment [30]. Most of the health care costs for treating low back pain are consumed by these patients [31]. It has been estimated that intractable, discogenic pain accounts for approximately 40% of patients with chronic low back pain [4].

The primary objective of this article is to provide a perspective and a rationale for the creation of specific ICD-10-CM codes that describe, characterize, and define pain associated with lumbar and lumbosacral degenerative disc disease. This proposal was developed under the aegis of the International Society for the Advancement of Spine Surgery (ISASS) with the endorsement of the American Academy of Orthopaedic Surgeons (AAOS) and American Association of Neurological Surgeons (AANS).

## 2. Pathoetiology of Discogenic Pain

The intervertebral disc is a key, non-osseous structural element in the spine that serves as a critical load transfer junction at each vertebral motion segment and supports the complex, three-dimensional kinematics of the normal spine [32,33,34]. Composed of an inner gelatinous nucleus pulposus and a fibrous outer annulus, this unique morphology and structure reflects these functions. In the healthy disc, intact and abundant aggrecan, a major proteoglycan in articular cartilage, endows the disc with substantial load-bearing capacity and inhibits nerve and vascular ingrowth [35]. Consequently, for the disc to perform its biomechanical function most efficiently, it is commensurately aneural and avascular in the healthy, young spine [36,37].

Degeneration of the lumbar intervertebral disc precedes the degeneration of all other connective tissues in the human body, with imaging evidence often apparent by the end of the second decade [15,38,39,40]. As degeneration progresses, the intervertebral disc becomes less efficient in its ability to absorb physiological loads, since the nucleus, normally highly hydrated and rich in proteoglycans, loses its capacity to bind water under compression as collagen fibers become disorganized [41]. Diminishing pressure within the nucleus invariably leads to reduced disc height (Figure 1) [18]. This degenerative cascade results in aberrant load transfer to adjacent vertebral bodies which is antecedent to endplate changes, osteophyte formation, and trabecular microfractures [39]. Alterations in normal loading patterns resulting from disc degeneration are also borne by the facet joints, leading to arthrosis, hypertrophy, and possible compression of neural elements [15,42]. Indeed, a recent evidence synthesis suggests a strong pathophysiological interdependence across the entire three-joint complex, with the cascade of arthritic degeneration commencing in the disc and eventually spreading to the facet joints [7].

A large and growing body of published evidence has established that the intervertebral disc can be a potential pain generator, leading to diminished quality of life for millions of patients worldwide [4,8,11,13,17,18,20,22,37,43,44,45,46,47]. Pathologic internal disc disruption, characterized by degradation of the nuclear matrix and definitive annular tears or ruptures, is distinct from asymptomatic disc degeneration, a nearly universal phenomenon with aging in humans [6,8,22,42,48,49]. With this condition, the external appearance of the disc remains essentially normal, but the interior architecture undergoes disruption (Figure 2) [50]. Symptomatic disc degeneration is postulated to result from repetitive mechanical loading causing endplate microfractures and the disruption of outer annular fibers, resulting in a vicious cycle of matrix damage, persistent inflammation, the sensitization of developed annular nociceptors, and hypermobility [8,41,49,51,52,53,54,55,56].

The pathognomonic process of symptomatic internal disc disruption has evolved and is now more completely understood [3,4,8,10,20,37,57]. Imaging evidence of disc disease often includes a decreased disc signal in conjunction with a high-intensity zone on T2-weighted magnetic resonance images. Corresponding anatomical changes involve subchondral bone sclerosis, disc space narrowing or collapse, as well as posterior annular tears, fissuring, ruptures, and delamination, which can ultimately result in intractable low back pain [44,58,59,60]. Histological studies have demonstrated that disc degeneration and lamellar disruption are associated with neovascularization, neuronal penetration with unmyelinated nerve fibers, and the ingrowth of Schwann cells [22,49,61,62,63,64,65,66,67].

With radial annular fissures, nucleus pulposus matrix substance migrates to the outer annulus and induces nerve ingrowth into delaminated regions [68]. With solitary concentric annular fissures, the lesions involve torn collagen fibers that attract an inflammatory repair response [3]. Radial fissures that extend into the outer innervated third of the annulus correlate strongly with reproduction of the patient’s pain by discography, and are independent of age and degenerative changes [69,70]. In many cases, these radial fissures may be in direct communication with concentric tears most often located in the posterior or posterolateral annulus. These annular fissures are mechanically and chemically conducive to the ingrowth of nerves and blood vessels by providing a low-pressure microenvironment that facilitates focal proteoglycan loss, leaving a matrix that is receptive to nerve and blood vessel ingrowth [56]. In fact, biochemical evidence supports the premise that painful discs result from a repetitive cycle of injury and repair with vascularized granulation tissue becoming embedded along torn annular fissures [10,49,50,53]. Along these zones of granulation tissue, inflammatory mediators sensitize damaged annular regions to mechanical and chemical stimuli [19,20,37,58,62,71,72].

It is postulated that the ingrowth of nerve endings serves as the pathoanatomic correlate to the dull chronic ache experienced by patients with chronic low back pain often referred to as discogenic pain [73]. This type of pain is exacerbated by mechanical loading of the spine, particularly in flexion [28], which involves separation of the fibers of the posterior annulus as they separate from their sites of attachment [74]. The likelihood of a concentric or circumferential tear of the posterior or posterolateral annulus is increased if axial rotation occurs in combination with flexion. Furthermore, in flexion, the articular processes of the facet joints are subluxated, affording less resistance to axial rotation. As a result, during flexion–rotation, the annulus fibrosus is subjected to maximal biomechanical stresses, while least protected by the posterior elements. Consequently, in the case of a concentric annular tear, the pain will be aggravated by any movements that stress the annulus, particularly flexion and rotation in the same direction that produced the lesion [3]. The same pain mechanism can also be postulated for radial tears of the annulus, particularly when these lesions are coupled with concentric tears.

Approximately 40% of patients with persistent symptoms of chronic low back pain have evidence of internal disc disruption [17,75]. If back symptoms persist beyond 3 months, the vast majority of patients with definitive imaging evidence of internal disc disruption do not have a good prognosis for recovery with conservative management alone [30,76,77]. These patients may face possible opioid dependency or major surgery [28,78,79].

## 3. Clinical Presentation

The clinical diagnosis of lumbar discogenic disease is characterized by axial midline low back pain, sitting intolerance, pain with flexion, positive provocation with sustained hip flexion, absence of motor/sensor/reflex change, and positive discography (Figure 3) [4,80,81,82,83,84,85]. Discogenic pain is often described as dull, aching, and gnawing [81]. Although it is localized to the low back, it can also be coupled with referred pain to the lower extremities, usually above the knees in a non-dermatomal distribution. With referred leg pain, patients describe a deep tissue pain resulting from a sensation of expanding pressure [81]. Discogenic pain that involves the legs is always somatic in nature and sclerotomal in distribution as it expands into wide areas that can be difficult to localize (Figure 4). While the boundaries of the pain can be difficult to define, patients can confidently identify its center or core. Since somatic referred pain is not caused by the compression of nerve roots, there are no neurological radicular signs.

Consequently, the physical signs on clinical examination are distinctly different than pain associated with neurocompressive disorders. For example, in contrast to discogenic pain, low back pain related to disc herniation or spinal stenosis with radiculopathy can be identified by positive straight leg raise, Lasegue’s sign, crossed Lasegue’s sign, positive bowstring, positive femoral stretch tests, and motor/sensory/reflex change, as well as possible changes on electromyography. Lower extremity radicular pain most often presents as a thin band of pain with a lancinating quality that is described as electric and shocking [81].

## 4. Diagnostic Criteria

In conjunction with the specific signs and symptoms identified by physical examination, discography can provide a definitive test for confirming the diagnosis of disc degeneration associated with chronically severe discogenic pain [4,44,75,79,86,87,88,89,90,91,92]. Developed in the 1950s, discography is a minimally invasive diagnostic test that is most commonly performed in a provocative manner via the intradiscal injection of contrast agents to reproduce (i.e., diagnostic discogram) the patient’s pain. Additionally, the test can also be conducted using a very small dose of an anesthetic agent to relieve the pain (i.e., functional anesthetic discogram) [93].

This simple clinical test is not only essential prior to intradiscal treatments, but more importantly, it has been used for decades to confirm a diagnosis of discogenic pain and differentiating it from other pain generators. Provocative discography is often combined with computed tomography (i.e., CT discogram) to identify the location and extent of annular disruption. Posterior and/or posterolateral annular radial and/or concentric fissure(s) that extend to the outer annular fibers (e.g., grade 3 out of 4 severity) account for over 70% of painful discs [94]. The false-positive rate for low-pressure (<20 psi) discography has been reported to be as low as 6% [92].

Recently, it has been shown that a specialized magnetic resonance spectroscopy exam can identify chemically painful lumbar discs and is associated with clinically significant improvements in patient-reported outcomes following surgery for discogenic low back pain [95]. This advancement could provide a valuable new diagnostic tool to assist clinicians in better selecting patients and treatment levels, with the best chance of realizing successful and durable pain management.

## 5. Refining the Definition of Discogenic Pain: A Proposal

The objective of the ISASS proposal, supported by both AAOS and AANS, is to support the creation of new ICD-10-CM diagnosis codes for describing pain associated with lumbar and lumbosacral degenerative disc disease or lumbar “discogenic” disease. Currently, there is no specific code to classify and define discogenic pain as a unique source of pain distinct from other recognized sources of chronic low back pain including facetogenic, neurocompressive including herniation and/or stenosis, sacroiliac, vertebrogenic, and psychogenic. All of these other sources have well-defined ICD-10-CM codes. Corresponding codes for discogenic pain remain absent from the diagnostic coding vernacular.

ISASS proposes modernized ICD-10-CM codes that specifically define pain associated with lumbar and lumbosacral degenerative disc disease. The proposed codes would also allow the pain to be characterized by location: lumbar region only, leg only, or both. Overall, having specific codes will improve coding precision for lumbar degenerative disc disease with and without referred leg pain and, thus, lead to better clinical assessment, diagnosis, and treatment strategies.

## 6. Conclusions

Over many decades of intensive investigation into the etiology and pathophysiology of chronic low back pain, an enormous research repository has been amassed to definitively identify the intervertebral disc as a distinct and relatively common pain generator [4,8]. However, physicians continue to utilize a variety of diagnostic labels that lack specificity, clarity, and granularity for characterizing lumbar degenerative disc disease. We believe much of this confusion is due to the lack of specific ICD-10-CM codes to capture pain of discogenic origin. Indeed, because the existing ICD-10-CM terminology is both non-specific and/or out-of-date for lumbar disc degeneration manifesting as low back pain and/or referred leg pain, many of the services provided by medical personnel to manage these patients get identified by payers as either (1) investigational or (2) non-covered. Consequently, data collection efforts supporting these services are often flawed and cases are under-reported.

Recently, vertebrogenic pain was granted a specific ICD-10-CM diagnostic code. The new diagnosis code (M54.51) went into effect on 1 October 2021 and can be applied, for example, to patients meeting indications for treatment with basivertebral nerve radiofrequency neurotomy. It is worth noting that this diagnostic code is for the lumbar spine only. This same anatomical distinction to the lumbar spine could also apply to discogenic back pain.

Additionally, the newly granted ICD-10-CM code for vertebrogenic pain, “pain coming from endplate bone” (transmitted via the basivertebral nerve), couples the symptoms with the etiology. The association between lumbar disc degeneration and chronic low back pain has been well established and warrants similar coding schema for “pain coming from the disc” (annulus fibrosis pain transmitted via the sinuvertebral nerve).

## Figures and Tables

**Figure 1 jpm-13-00821-f001:**
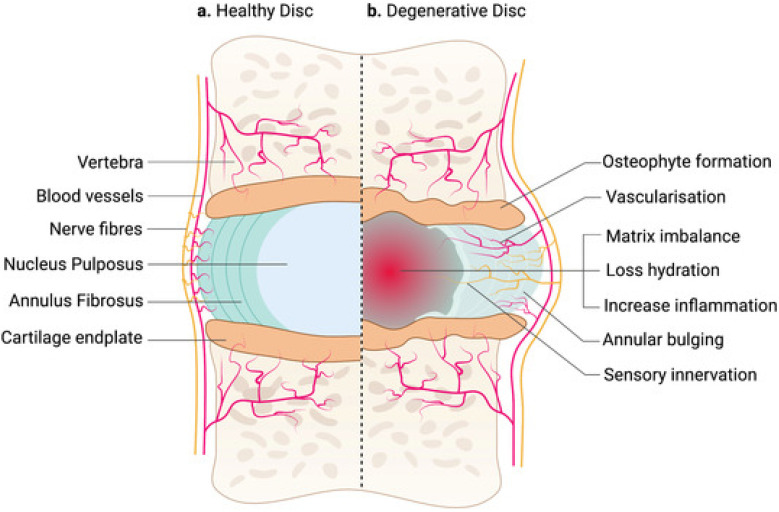
Side-by-side comparison of a healthy (**a**) and degenerated (**b**) intervertebral disc [18].

**Figure 2 jpm-13-00821-f002:**
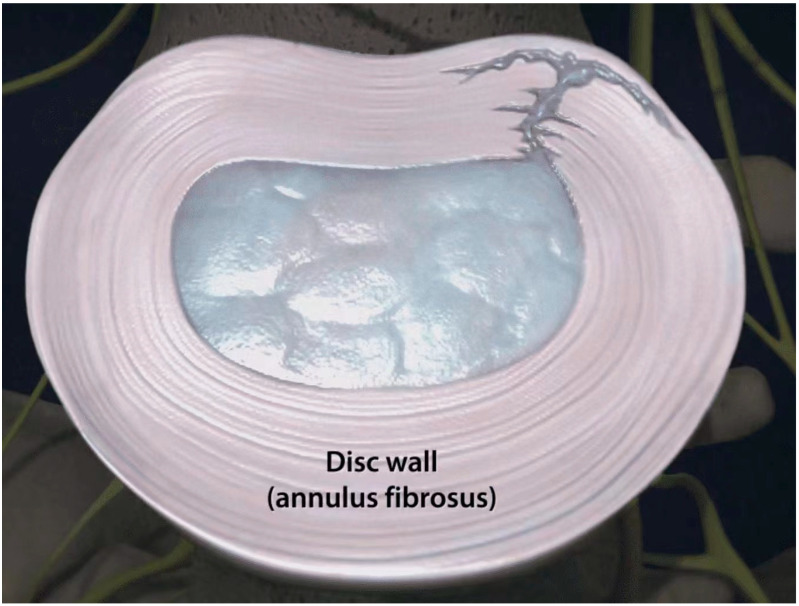
Combined radial and concentric tears extending to the outer layers of the posterolateral intervertebral disc annulus.

**Figure 3 jpm-13-00821-f003:**
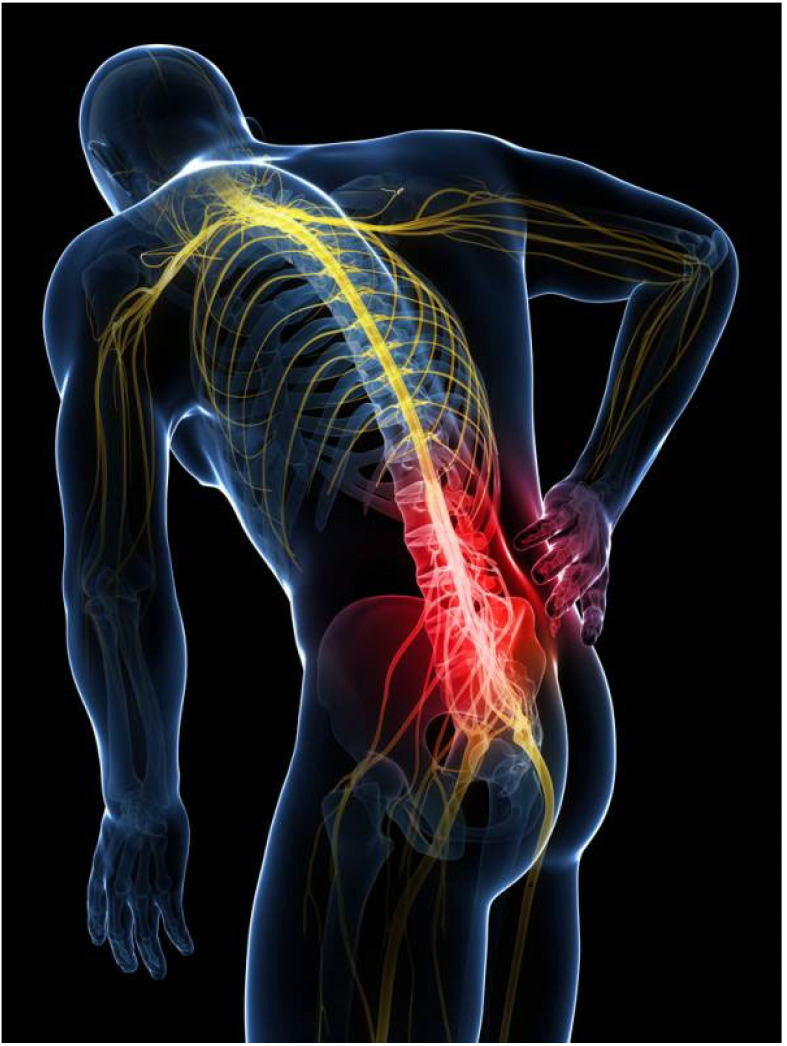
Axial midline low back pain.

**Figure 4 jpm-13-00821-f004:**
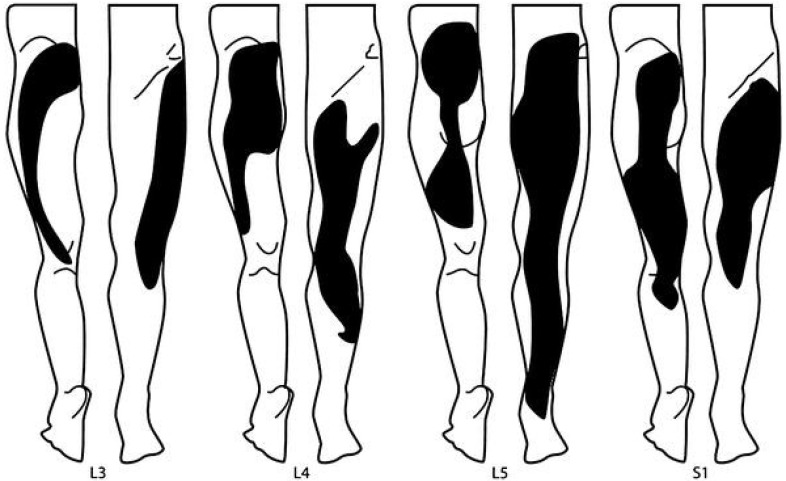
Sclerotomal pain distribution patterns in referred leg pain of discogenic origin by vertebral level (Adapted from Kellgren, J.H., Clin Sci 1939; 4: 35–46).

## Data Availability

Not applicable.

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
