# Peer review of "Defining the Patient with Lumbar Discogenic Pain: Real-World Implications for Diagnosis and Effective Clinical Management"

_jpm, 2023, doi:10.3390/jpm13050821_

Round 1
Reviewer 1 Report
I would suggest changing the first statement's references and picking the most relevant 1-5 most relevant and up to date references to support to statement instead of 62 articles.
Other than that I think its an interesting article from the data aspect and as a general review of discogenic pain for medical professionals and a well written one.
Author Response
Response to Reviewer 1
I would suggest changing the first statement's references and picking the most relevant 1-5 most relevant and up to date references to support to statement instead of 62 articles.
Per the reviewer’s request, we have significantly reduced the number of citations in support of the opening statement.
Other than that I think it’s an interesting article from the data aspect and as a general review of discogenic pain for medical professionals and a well written one.
Much appreciated.
Reviewer 2 Report
The present study underlines the absence of a specific diagnostic code for lumbar discogenic pain in the ICD-10-CM. For this reason, the Authors reported some clinical and radiological findings for discogenic pain.
Even if the topic is interesting, some points should be improved:
- In the real world, the main problematic implication for discogenic pain is not the absence of a specific code but the difficulty in making a correct diagnosis. In fact, even if discography is suggested as a gold standard, the sensibility and specificity of this procedure is sometimes not appropriate for a diagnosis. On the contrary, from a radiological point of view, HIZ of the anulus has low specificity. The Authors should underline and discuss the difficulty in making a correct diagnosis of discogenic pain, along with the correct technique for performing discography. They should also clarify that discogenic pain is frequently a diagnosis of exclusion in patients with low back pain.
Author Response
Response to Reviewer 2
The present study underlines the absence of a specific diagnostic code for lumbar discogenic pain in the ICD-10-CM. For this reason, the Authors reported some clinical and radiological findings for discogenic pain.
Even if the topic is interesting, some points should be improved:
- In the real world, the main problematic implication for discogenic pain is not the absence of a specific code but the difficulty in making a correct diagnosis. In fact, even if discography is suggested as a gold standard, the sensibility and specificity of this procedure is sometimes not appropriate for a diagnosis. On the contrary, from a radiological point of view, HIZ of the anulus has low specificity. The Authors should underline and discuss the difficulty in making a correct diagnosis of discogenic pain, along with the correct technique for performing discography. They should also clarify that discogenic pain is frequently a diagnosis of exclusion in patients with low back pain.
To address this important issue, we have added text to section 4 (Diagnostic Criteria) indicating that provocative discography in conjunction with specific physical exam characteristics detailed in section 3 (Clinical Presentation) are required to confirm a diagnosis of disc degeneration as the source of pain.
Additionally, we have included a recently published citation (Gornet et al) showing that there were more successful, sustained clinical outcomes associated with surgically treating chemically painful discs identified by a specialized magnetic resonance spectroscopy exam. This could provide a valuable new diagnostic tool to help clinicians better select treatment levels.